# In Vitro Evaluation of Gentamicin or Vancomycin Containing Bone Graft Substitute in the Prevention of Orthopedic Implant-Related Infections

**DOI:** 10.3390/ijms21239250

**Published:** 2020-12-04

**Authors:** Alessandro Bidossi, Marta Bottagisio, Nicola Logoluso, Elena De Vecchi

**Affiliations:** 1IRCCS Istituto Ortopedico Galeazzi, Laboratory of Clinical Chemistry and Microbiology, 20161 Milan, Italy; alessandro.bidossi@grupposandonato.it (A.B.); elena.devecchi@grupposandonato.it (E.D.V.); 2IRCCS Istituto Ortopedico Galeazzi, Department of Reconstructive Surgery of Osteo-Articular Infections C.R.I.O. Unit, 20161 Milan, Italy; nicola.logoluso@grupposandonato.it

**Keywords:** prosthetic joint infection, fracture-related infection, bone graft substitute, revision arthroplasty, local antibiotic

## Abstract

Antibiotic-loaded bone graft substitutes are attractive clinical options and have been used for years either for prophylaxis or therapy for periprosthetic and fracture-related infections. Calcium sulfate and hydroxyapatite can be combined in an injectable and moldable bone graft substitute that provides dead space management with local release of high concentrations of antibiotics in a one-stage approach. With the aim to test preventive strategies against bone infections, a commercial hydroxyapatite/calcium sulfate bone graft substitute containing either gentamicin or vancomycin was tested against *Staphylococcus aureus*, *Staphylococcus epidermidis* and *Pseudomonas aeruginosa*, harboring different resistance determinants. The prevention of bacterial colonization and biofilm development by selected microorganisms was investigated along with the capability of the eluted antibiotics to select for antibiotic resistance. The addition of antibiotics drastically affected the ability of the selected strains to adhere to the tested compound. Furthermore, both the antibiotics eluted by the bone graft substitutes were able to negatively impair the biofilm maturation of all the staphylococcal strains. As expected, *P. aeruginosa* was significantly affected only by the gentamicin containing bone graft substitutes. Finally, the prolonged exposure to antibiotic-containing sulfate/hydroxyapatite discs did not lead to any stable or transient adaptations in either of the tested bacterial strains. No signs of the development of antibiotic resistance were found, which confirms the safety of this strategy for the prevention of infection in orthopedic surgery.

## 1. Introduction

In the 1960s, the first total knee and hip replacements were implanted in patients, heralding the starting point of a revolution in orthopedic surgery [1]. Today, total joint replacement and internal fixation of fractures represent two of the most successful interventions in modern medicine, improving the quality of life of and fostering longevity in an elderly population. The rising number of orthopedic implants is unfortunately associated with an increase in implant-related infections (IRI), which represents a significant clinical, public and economic burden. IRI may be divided into periprosthetic joint infections (PJI) [2,3,4] and fracture-related infections (FRI) [5,6], both presenting two different clinical and surgical challenges.

The most commonly isolated bacteria causing IRI are *Staphylococcus aureus* and coagulase-negative staphylococci, which account for more than 60% of PJI, followed by other clinically relevant microbes such as streptococci, enterococci, Gram-negative rods and anaerobes [7]. Most of them can form a biofilm, which is defined as an aggregate of microbial cells embedded within a slimy self-produced matrix [8]. Single free-floating bacterial cells start aggregating and attaching to implant surfaces. An immature biofilm continues growing in cellular density and, later on, extracellular components promote microbial aggregation and stimulate the development of the slimy extracellular matrix, conferring extreme resistance to the action of immune cells and antimicrobials [9]. In addition, bacterial resistance and tolerance to antimicrobial agents [10] play a key role in infection persistence and therapeutic failure [11]. The cornerstones of effective eradication of the bacterial IRI are the resection of necrotic tissue by surgical debridement and antibiotic therapy [12,13]. Systemic antibiotic therapy, however, often offers only low local concentrations, depending on bioavailability and tissue penetration, and can cause severe adverse events. The additional use of local antibiotics can achieve high doses of active drugs exceeding the minimum inhibitory concentrations (MICs) [11] exclusively at the site of infection, thereby optimizing therapeutic efficacy and minimizing the risk of systemic toxicity [14].

Different degradable biomaterials have been suggested as antibiotic carriers [15]. Specifically, antibiotic-containing bone graft substitutes (BGS) were designed to promote and protect the bone healing process. CERAMENT^TM^ is a resorbable biphasic biomaterial with a powder composition of 40% hydroxyapatite and 60% calcium sulfate, which offers a specific balance between material dissolution and bone regeneration. Premanufactured with either gentamicin (CERAMENT^TM^|G) or vancomycin (CERAMENT^TM^|V), it is aimed at ensuring antibacterial protection during the bone regeneration process.

Pre-clinical and clinical investigations of the prefabricated gentamicin-eluting BGS reported a prolonged release of high drug concentrations [16,17,18], which demonstrated that it was effective in vitro against sessile bacteria [19]. The clinical impact of the BGS has been reported in the literature [20,21,22], proving its efficacy and safety in the management of orthopedic infections. However, in vitro data concerning the activity of the vancomycin-loaded BGS are needed to support the clinical use of this formulation. Hence, the aim of this study was to further investigate in vitro the ability of the antibiotic-eluting hydroxyapatite/calcium sulfate BGS to prevent bacterial adhesion and biofilm formation by clinically relevant microorganisms, which were isolated from PJI and FRI, together with the evaluation of a possible selection for bacterial resistance to the eluted antibiotics.

## 2. Results

### 2.1. Bacterial Adhesion on Material Surface

#### 2.1.1. *Staphylococcus aureus*

Results observed when testing *S. aureus* adhesion on the different materials are summarized in Figure 1a. The clinical isolate susceptible to all antibiotics (SauS) showed greater affinity to the surface of unloaded ceramic bone void filler (CBVF) than to sandblasted titanium (ST), with a significant increase in the number of attached cells concordant with the incubation time.

After incubation for 120 min, 24 ± 25.3 CFU/mL were found on ST vs. 73.3 ± 42.5 CFU/mL on CBVF. Ceramic filler containing gentamicin (CG) and vancomycin (CV) completely inhibited the adhesion of SauS at all time points. Different from SauS, the glycopeptide-intermediate *S. aureus* (GISA) strain showed no significant difference in the adhesion to ST or CBVF. At 120 min, 48 ± 45.8 CFU/mL were found on ST vs. 38.6 ± 27.4 CFU/mL on CBVF. CG and CV were still able to prevent the adhesion of GISA at all time points completely (0 CFU/mL).

At 120 min, ST exposed a significantly higher attachment of gentamicin-resistant *S. aureus* (SauG) than CBVF or CG (857.3 ± 211 vs. 242.6 ± 92.5 vs. 320 ±178.4 CFU/mL, respectively). Indeed, CG could not inhibit the adhesion of the gentamicin-resistant SauG (300 CFU/mL); only CV was able to completely impede SauG adhesion (0 CFU/mL).

In contrast to all other strains, methicillin-resistant *S. aureus* (MRSA) showed a higher adhesion to CG with respect to ST and CBVF (96 ± 37.1 vs. 29.3 ± 28.1 vs. 48 ± 37.6 CFU/mL, respectively) at 120 min. As observed with the other staphylococcal strains, CV completely inhibited MRSA adhesion (0 CFU/mL).

#### 2.1.2. *Staphylococcus epidermidis*

The *S. epidermidis*-susceptible strain (SepS), the gentamicin-resistant strain (SepG) and methicillin-resistant *S. epidermidis* (MRSE) showed a significantly higher adhesion to ST than to CBVF after 120 min (Figure 1b). In glycopeptide-intermediate clinical isolate (GISE) no significant differences in the adhesion to ST or CBVF were found.

Similar to the observation on *S. aureus*, CG and CV were able to inhibit the adhesion of SepS and GISA at all time points (0 CFU/mL). Interestingly, CG significantly impaired the adhesion of the resistant SepG if compared with CBVF (28 ± 21.1 vs. 153.3 ± 36 CFU/mL) and even completely inhibited the adhesion of MRSE at all time points. CV was able to prevent the colonization of the disc’s surface of all the tested *S. epidermidis* strains, regardless of the antibiotic susceptibility profile.

#### 2.1.3. *Pseudomonas aeruginosa*

Both the gentamicin-susceptible (PaeS) and the gentamicin-resistant (PaeG) *P. aeruginosa* strains demonstrated a higher surface affinity to CBVF than to ST, regardless of their antibiotic susceptibility profile (Figure 1c). In comparison to CBVF, CV was able to significantly decrease the adhesion of PaeS (150.6 ± 74 vs. 350.6 ± 91.9 CFU/mL at 120 min) and PaeG (22.6 ± 19.8 vs. 118.6 ± 66.5 CFU/mL at 120 min) at all time points. CG was able to completely prevent the adhesion not only of PaeS, but also of the gentamicin-resistant PaeG at all time points.

### 2.2. Biofilm Formation

In the analysis of biofilm formation by confocal laser scan microscopy (CLSM), all tested bacterial isolates demonstrated a significantly higher biofilm biomass production on ST than on CBVF (Figure 2). Specifically, the biofilm biomass was 9.0 × 10^7^ ± 3.9 × 10^6^ µm^3^ on ST vs. 9.2 × 10^7^ ± 2.0 × 10^6^ µm^3^ on CBVF for MRSA, 6.8 × 10^7^ ± 2.3 × 10^7^ µm^3^ on ST vs. 1.6 × 10^7^ ± 5.2 × 10^6^ µm^3^ on CBVF for MRSE and 3.5 × 10^7^ ± 1.2 × 10^6^ µm^3^ on ST vs. 5.2 × 10^6^ ± 4.5 × 10^5^ µm^3^ on CBVF for PaeG, respectively. Representative 3D reconstructions of mature biofilm and surface structures of the analyzed substrates are depicted in Figure 3 and Figure 4.

#### 2.2.1. *Staphylococcus aureus*

CSLM analysis allowed quantification of the biofilm biomass at ~6.2 × 10^6^ ± 1.2 × 10^6^ µm^3^ on CBVF for SauS (Figure 5a). The biofilm biomass was significantly reduced to ~1.2 × 10^6^ ± 7.4 × 10^4^ µm^3^ on CG and to ~1.4 × 10^6^ ± 3.9 × 10^5^ µm^3^ on CV. In MRSA and GISA, similar patterns were found with a significant reduction of the biofilm biomass on CG or CV. For SauG, only CV was able to significantly decrease the biofilm production, whereas CG did not show a significant reduction. However, as reported in Table 1 and depicted in Figure 5a, CG displayed a markedly increased mortality rate of MRSA and GISA, as compared to CV (55.3% and 48.9% vs. 38.6% and 20.6%, respectively).

#### 2.2.2. *Staphylococcus epidermidis*

The CLSM results for the *S. epidermidis* strains are comparable to the results observed for *S. aureus*. SepS, GISE, SepG and MRSE demonstrated a significantly lower biofilm-forming capacity on CG or CV, as compared to CBVF.

For example, MRSE biofilm biomass was ~1.6 × 10^7^ ± 5.2 × 10^6^ µm^3^ on CBVF, ~1.8 × 10^6^ ± 2.7 × 10^5^ µm^3^ on CG and ~2.2 × 10^6^ ± 3.7 × 10^5^ µm^3^ on CV, respectively. Notably, CG was able to significantly hinder biofilm formation even in the gentamicin-resistant SepG (1.1 × 10^7^ ± 1.2 × 10^6^ µm^3^ on CBVF vs. 3.6 × 10^6^ ± 1.4 × 10^6^ µm^3^ on CG). When considering the bactericidal activity against sessile cells, CG displayed an increased mortality rate of the MRSE and GISE strains, similar to that observed for *S. aureus*, whereas CV induced a notable reduction of vital cells towards SepG (Table 1, Figure 5b).

#### 2.2.3. *Pseudomonas aeruginosa*

CV was not able to suppress biofilm development of PaeS and PaeG (Figure 5c). Biofilm biomass of PaeS was 5.1 × 10^6^ ± 1.6 × 10^6^ µm^3^ on CBVF vs. 5.3 × 10^6^ ± 4.2 × 10^5^ µm^3^ on CV. However, CV seemingly induced a deeper stress when compared to CBVF, since the sessile biomass of both *P. aeruginosa* strains displayed an increased percentage of dead cells (Table 1, Figure 5c). As expected, CG displayed greater activity in preventing *P. aeruginosa* biofilm formation by significantly reducing the biomass of both isolates, being more effective against PaeS (5.1 × 10^6^ ± 1.6 × 10^6^ µm^3^ on CBVF vs. 2.1 × 10^6^ ± 2.9 × 10^5^ µm^3^ on CG) than against PaeG (5.2 × 10^6^ ± 4.5 × 10^5^ µm^3^ on CBVF vs. 2.7 × 10^6^ ± 1.7 × 10^5^ µm^3^ on CG).

### 2.3. Resistance Selection to Eluted Vancomycin and Gentamicin

In the tested experimental conditions, no adaptation in the presence of CG or CV was observed by means of the zone of inhibition (ZOI) measurements on a solid agar medium. When repeatedly exposed to a freshly prepared antibiotic-loaded disc, all the tested strains displayed uniform ZOIs with no colonies of viable bacteria inside the inhibition halo.

When in the presence of a new disc, all the strains of *S. aureus* and *P. aeruginosa* maintained a comparable susceptibility profile (ZOI diameter) at all the time points, as depicted in Figure 6a,c, respectively (∆ ZOI = ± 2 mm between T1 and T7), with the exception of the MRSA strain, which showed a slight decrease in susceptibility to gentamicin after seven consecutive cycles (ZOI 10.3 ± 0.6 mm and 4.3 ± 0.6 mm at T1 and T7, respectively). In contrast, *S. epidermidis* strains, except for SepG, displayed an increased ZOI diameter after the second CG cycle (14.7 ± 0.6 mm vs. 19.3 ± 0.6 mm for SepS, 14.3 ± 0.6 mm vs. 17.3 ± 1.2 mm for MRSE and 14.3 ± 0.6 mm vs. 17 ± 1 mm for GISE at T1 and T2, respectively) (Figure 6b). Nevertheless, the increased susceptibility profile proved to be transitory as the growth–inhibition diameters reduced during the following cycles, reaching values like those observed at T1 (∆ ZOI = ± 2 mm between T1 and T7). All other differences observed were not significant.

When the same disc was used from T1 to T7, all the tested staphylococcal strains showed an absence of inhibition halo at the later time points in the presence of CV, comprising both the glycopeptide-intermediate strains (GISA and GISE). Otherwise, in the presence of CG, the gentamicin-resistant strains (SauG and SepG) were not inhibited from T4 and T3, respectively. The absence of any stable or transient adaptation was then confirmed by measuring MIC values for all the tested clinical isolates exposed to both gentamicin and vancomycin released from antibiotic-loaded discs. No significant changes in the susceptibility profile were observed. Indeed, MIC fluctuations recorded never exceeded 1-fold dilution (Table 2 and Table 3).

## 3. Discussion

One of the most feared complications in orthopedic surgery is the bacterial colonization of abiotic surfaces causing PJI and FRI, occurring in 1% to 2% of the patients receiving orthopedic implants [23]. Antibiotic-eluting BGS represent an attractive clinical option in the management of PJI and FRI. Gentamicin-loaded ceramic fillers have been extensively employed in clinics and their successful use reported in the literature [20,21,22,24,25,26,27,28]. The increasing emergence of aminoglycoside resistance has led to the development of a new BGS formulation enriched with vancomycin, an additional option alongside gentamicin in the orthopedic field [29]. Although these are promising strategies, there is a lack of in vitro data supporting the safe use of both CG and CV in the treatment of biofilm-related infections. Indeed, their impact on bacterial adhesion, biofilm formation and selection of resistance has not been systematically evaluated in vitro.

### 3.1. Bacterial Adhesion

Clinically relevant strains of *S. aureus*, *S. epidermidis* and *P. aeruginosa* were tested for their ability to adhere in the first two hours of contact to solidified calcium sulfate/hydroxyapatite discs, to evaluate a possible contamination upon surgical implantation. This scenario can be expected in a contaminated open fracture (e.g., Gustilo–Anderson IIIb), where, even after the initial extensive debridement and irrigation, bacteria may persist and attach to bone fragments or internal fixation devices. When adhesion to unloaded CBVF was compared to ST, staphylococcal isolates displayed a heterogeneous trend, which was mainly similar or lower to that of ST, with the exception of the two susceptible strains. On the contrary, all the *P. aeruginosa* isolates showed a greater affinity for CBVF discs than for ST under the tested experimental conditions. It is well known that this reversible event not only varies according to the physical characteristics of the implant, but also relies on differences between bacterial species and even strains. Indeed, the surface hydrophilicity of the material, the presence of functional groups and electric charge, as well as the physicochemical characteristics of the microbial membrane modulate initial physicochemical interactions influencing early bacterial adhesion [30]. In this context, the use of antibiotic-loaded CG or CV disks drastically affected the ability of the selected strains to adhere to the tested compounds. CG inhibited the adhesion of the gentamicin-susceptible bacterial strains (SauS, GISA, SepS and GISE) and reduced the bacterial load of the gentamicin-resistant staphylococci (SauG, SepG, MRSA and MRSE), but could not completely impede their adhesion. Since gentamicin is active against Gram-negative bacteria, CG inhibited the attachment of PaeS and PaeG, whereas CV completely avoided the adhesion of all staphylococci strains. As expected, CV was not able to impair the adhesion of the Gram-negative *P. aeruginosa* strains.

In the clinical setting, this finding could be used in the prophylaxis of infection in contaminated open fractures. If a bone void or fracture gap is present during the initial debridement, it could be filled with CG. Since open fractures might be contaminated by Gram-positive and Gram-negative bacteria [31], the shown effect of CG on *P. aeruginosa* might be beneficial in this setting. Aljawadi and co-workers recently presented their clinical results on the use of CG in addition to systemic antibiotics in Gustilo–Anderson IIIb open fractures in a single-stage “Fix and Flap” procedure in 80 patients [24]. The limb salvage rate in this challenging patient group was 96.25% and the infection rate was 1.25%. A randomized controlled trial for CG as part of the surgical repair for open tibial diaphyseal fractures is on its way and recruitment of 200 patients has been finalized (www.clincaltrials.gov, Identifier: NCT02820363).

### 3.2. Evaluation of Biofilm Formation

The ability to form biofilm on the tested substrates was evaluated by CLSM. Despite the porous structure of CBVF, all the tested strains displayed a significantly greater biomass volume on sandblasted titanium than on CBVF.

Interestingly, biofilm formation by all the tested staphylococcal strains was negatively affected by the presence of antibiotic-eluting BGS, highlighting the efficacy of the initial burst release of an antimicrobial even in such an in vitro setting conducted under ideal conditions, which does not accurately recapitulate in vivo conditions. This experimental design does not consider important aspects such as the lack of surrounding tissues, the physiological supply of nutrients, cellular resorption and immune system action, which might concur in impairing bacterial contamination and help to reduce the number of bacteria on the site of contamination. The high bacterial inoculum (much greater than that of a possible contamination event, which is likely to happen from few cells) and a nutrient-rich growth medium represent a “worst case scenario”, which understandably supports our results.

CG was able to significantly suppress biofilm formation of all staphylococci strains, except for SauG and both *P. aeruginosa* strains. CV reduced biofilm formation of all staphylococci, including MRSA and MRSE, but had no impact on Gram-negative *P. aeruginosa* biofilm biomass. The observed increase of mortality rate of cells encased within the biofilm matrix could be ascribed to the effects of the high vancomycin concentration, which might cause the perturbation of cell shape, as observed on Gram-negative strains by others [32].

To date, clinical outcomes of CG and CV have been published in PJI [22], FRI [25,27], chronic Osteomyelitis (cOM) [20,21], and Diabetic Foot Osteomyelitis (DFO) [26,28]. McNally and co-workers found a reduction of the recurrence rate of cOM of 4% in a one-stage procedure compared to common reports in the literature of about 10% [21]. Briefly, they evaluated the use of CG in the management of deep bone infection in a one-stage procedure. Considering the unknown bacterial resistance pattern, the broader Gram-negative and Gram-positive spectrum and the unavailability of CV at the time of the study, CG proved effective in the treatment of chronic Osteomyelitis, even in the presence of gentamicin-resistant clinical isolates.

The results of our in vitro investigation showed that CV might be beneficial in the management of staphylococcal infections. Selecting between CG and CV is only possible if the causative bacteria have been identified in the first stage of a two-stage procedure. Logoluso et al. used a two-stage approach in the management of PJI, where CG was used in 13 patients and CV in 7 (only staphylococcal and enterococcal infections), finding only one recurrence of PJI in their series of 20 consecutive cases [22]. In FRI, the recurrence rate was reported in 7.7% [25] and 8.5% of cases [27], and in DFO in 10% [26].

### 3.3. Resistance Selection by Eluted Vancomycin and Gentamicin

There is an ongoing discussion if local antibiotics can induce resistance or lead to a selection of resistant bacteria. The slow penetration rate of antibiotics through the biofilm matrix may give time to the cells to adapt phenotypically and genotypically. Sessile bacteria encased within the biomass are able to incorporate eDNA from the biofilm matrix, and in vitro tests suggested that horizontal gene transfer may even be 10,000 times more likely in biofilms compared to their planktonic counterparts [33,34,35,36]. Furthermore, cells in a biofilm are inherently prone to spontaneous mutations, possibly due to an increased response to stress and consequent DNA damage [37,38]. In the orthopedic field, this topic is gaining particular interest with regard to reduced staphylococcal susceptibility to glycopeptide antibiotics, which are commonly employed in the treatment of orthopedic infections because of their noteworthy bone penetration and the diffusion of methicillin-resistant strains [39]. While all ZOIs of gentamicin-resistant strains (SauG, SepG and PaeG) were notably inferior to those of non-resistant strains, such marked difference was not observed in the presence of vancomycin for the glycopeptide-intermediate strains. This result confirms that aminoglycoside resistance in staphylococci is derived from the acquisition of a gene (i.e., aph(3’)-IIIa), whereas resistance to glycopeptide mainly consists of a reduced susceptibility by accumulation of point mutations [40]. However, it is worth noting that glycopeptide-intermediate staphylococci behavior in the presence of CV was like that of the other strains.

For all the aforementioned reasons, it is extremely important to test whether the exposure to antibiotic-loaded biomaterials might lead to the acquisition of a resistance profile. The results obtained from the present study suggest that prolonged exposure either to the same or new antibiotic-eluting hydroxyapatite/calcium sulfate discs did not lead to stable or transient adaptations in either of the tested bacterial strains.

Despite the aforementioned limitations of the study, the emergence of antibiotic-resistant strains in the presence of an antibiotic-enriched compound in a clinical scenario seems extremely unlikely considering the initial burst of high concentrations of active drugs and the subsequent retention of a sustained amount of antibiotics, as also reported from previous studies [19]. Furthermore, the combined use of both gentamicin and vancomycin-eluting compounds might be considered for clinical cases presenting high-risk of infection recurrence due to polymicrobial contamination. This low-risk profile, together with the clinically confirmed regenerative properties of the tested resorbable materials [20,41], confirms further the pertinence of antibiotic-containing BGS as prophylactic and treatment options and proves them potentially safe for clinical application in orthopedic infection surgery [42].

## 4. Materials and Methods

### 4.1. Tested Bacterial Strains and Biomaterials

Clinically relevant bacterial strains from patients with IRI were isolated at the Laboratory of Clinical Chemistry and Microbiology of the IRCCS Galeazzi Orthopedic Institute and were used in this study. Biofilm-producing strains of *S. aureus* (one susceptible strain, SauS; one gentamicin-resistant strain, SauG; one methicillin-resistant strain, MRSA; one glycopeptide-intermediate strain, GISA), *S. epidermidis* (one susceptible strain, SepS; one gentamicin-resistant strain, SepG; one methicillin-resistant strain, MRSE; one glycopeptide-intermediate strain, GISE) and *P. aeruginosa* (one gentamicin-susceptible strain, PaeS and one gentamicin-resistant strain, PaeG) were selected. The bacterial ability to adhere and to produce biofilm, as well as the emergence of antibiotic resistance in vitro were tested on the following substrates: sandblasted titanium (ST; used as growth control) (Geass, Pozzuolo del Friuli, Italy), CERAMENT^TM^|BONE VOID FILLER, a biphasic degradable bone graft substitute composed of 40% hydroxyapatite and 60% calcium sulfate (without antibiotic, used as growth control), CERAMENT^TM^|G, containing 17.5 mg gentamicin/mL paste and CERAMENT^TM^|V, containing 66 mg vancomycin/mL paste. Sterile kits of the products were supplied by BONESUPPORT AB (Lund, Sweden) and mixed according to the instructions for use provided by the manufacturer.

### 4.2. Evaluation of Bacterial Adhesion on Material Surface

The effects of different formulations of the calcium sulfate/hydroxyapatite bone graft substitute on bacterial adhesion were tested. To standardize the experimental conditions, 6 × 2.5 mm discs of unloaded CBVF and CG and CV were produced by pouring freshly prepared injectable pastes into a custom-made mold of polydimethylsiloxane (PDMS, Farnell, Lainate, Italy). The theoretical concentration of gentamicin in each disc was approximately 1.23 mg, whereas that of vancomycin was approximately 4.66 mg per disc. Discs were placed in a 96-well microplate containing 180 μL of Brain Heart Infusion (BHI) broth (BioMérieux, Craponne, France). Each well was inoculated with 20 μL of a bacterial overnight culture to obtain a final density of 1.0 × 10^6^ CFU/mL. To evaluate the first phases of bacterial attachment upon tested substrates, after 30, 60 and 120 min of incubation at 37 °C, the discs were rinsed three times with sterile saline to remove non-adherent bacteria [43,44]. Then, discs were immersed in 200 μL of a 0.1% *w*/*v* dithiothreitol (DTT; Sigma-Aldrich, Milan, Italy) solution and mechanically stirred for 15 min at room temperature to detach bacteria adhering to the discs [45]. Finally, 10 μL from each well was serially diluted and drop-plated on Tryptic Soy Agar (TSA; Sigma-Aldrich, Milan, Italy) plates and incubated at 37 °C for 24 h for colony forming units (CFU) count [46]. The adhesion tests were performed in triplicates for each strain.

### 4.3. Analysis of Biofilm Formation by CLSM

The ability of the tested clinical strains to form biofilm on the antibiotic-containing and unloaded biphasic bone graft substitutes was evaluated using CLSM [47]. Discs were placed in a 24-well microplate with 950 μL of BHI broth (Biomérieux, Craponne, France) and 50 μL of 1.0 × 10^7^ CFU/mL bacterial suspension. After 48 h of incubation at 37 °C, discs were rinsed three times with sterile saline to remove non-adherent bacteria and stained with Filmtracer™ LIVE/DEAD™ Biofilm Viability Kit (Thermo Fisher Diagnostics, Monza, Italy), according to the manufacturer’s instructions. Briefly, the staining solution was prepared by adding 3 μL of SYTO9 and 3 μL of propidium iodide to 1 mL of filter-sterilized water. Samples were incubated with 20 μL of staining solution at room temperature in the dark for 15 min. Afterwards, samples were washed with sterile saline and examined with an upright TCS SP8 (Leica Microsystems CMS GmbH, Mannheim, Germany) using a 20× dry objective (HC PL FLUOTAR 20×/0.50 DRY). Images from at least three randomly selected areas were acquired for each sample. The obtained images were processed with Las X (Leica Microsystems CMS GmbH, Mannheim, Germany) and analyzed with Fiji software (Fiji, ImageJ, Wayne Rasband National Institutes of Health). The test was performed in triplicates for each strain.

### 4.4. Determination of Minimum Inhibitory Concentration

The MIC for each strain was determined by the broth microdilution method, following the guidelines of the European Committee on Antimicrobial Susceptibility Testing [48]. A microbial suspension was prepared for each bacterial strain in Mueller–Hinton (MHB, Millipore, Milan, Italy) broth to an optical density equal to a standard turbidity of 0.5 McFarland (~1.5 × 10^8^ CFU/mL). After obtaining a microbial load of 5 × 10^6^ CFU/mL, 10 μL of each suspension was inoculated in a 96-well microplate containing 90 μL of a serial 2-fold dilution of either gentamicin (Sigma-Aldrich, Milan, Italy) or vancomycin (VWR, Milan, Italy). The MIC was defined as the lowest antibiotic concentration that inhibited the bacterial growth after 24 h of incubation at 37 °C, as assessed by the unaided eye. The experiment was performed in triplicates.

### 4.5. Evaluation of Selection for Bacterial Resistance after Exposure to Eluted Antibiotics

The emergence of bacterial resistance to the antibiotics released from the drug-loaded formulations was investigated by the modified Kirby–Bauer method as reported by others [49]. Briefly, either gentamicin or vancomycin-eluting discs were placed on TSA plates previously seeded with a bacterial suspension corresponding to a standard turbidity of 0.5 McFarland (~5 × 10^8^ CFU/mL) (T0). After incubation at 37 °C for 24 h, the zone of inhibition (ZOI) was measured and bacterial cells growing on the edge thereof were subcultured on a fresh TSA plate for 7 days (from T1 to T7). Specifically, two different procedures were applied: (i) the same exhausted antibiotic-containing disc was used throughout the whole experiment (up to T7); (ii) the antibiotic-containing substrate was replaced with a fresh disc every 24 h for 7 days. The ZOI was measured and recorded daily. After overnight incubation in antibiotic-free BHI broth, bacteria were further subcultured on antibiotic-free TSA plates for an additional 7 days (T14) to assess the stability of the potentially acquired resistance. MIC values were determined before (T0) and after exposure to the eluted antibiotics (T7) and after the last cycle on antibiotic-free TSA plates (T14). The acquisition of bacterial resistance to an antimicrobial compound was defined as ≥4-fold increase in MIC values [50].

## 5. Conclusions

From the results obtained in this in vitro study, it can be concluded that CV efficacy is comparable to CG in the treatment of Gram-positive biofilm and has been proven to be effective against gentamicin-resistant strains. The use of antibiotic-containing hydroxyapatite/calcium sulfate bone graft substitutes could be a feasible strategy in the prevention of bacterial colonization and biofilm development by microorganisms and may prove to be a promising approach for dead-space management in the treatment of cOM, FRI, DFO and PJI, always combined with systemic antibiotic therapy.

## Figures and Tables

**Figure 1 ijms-21-09250-f001:**
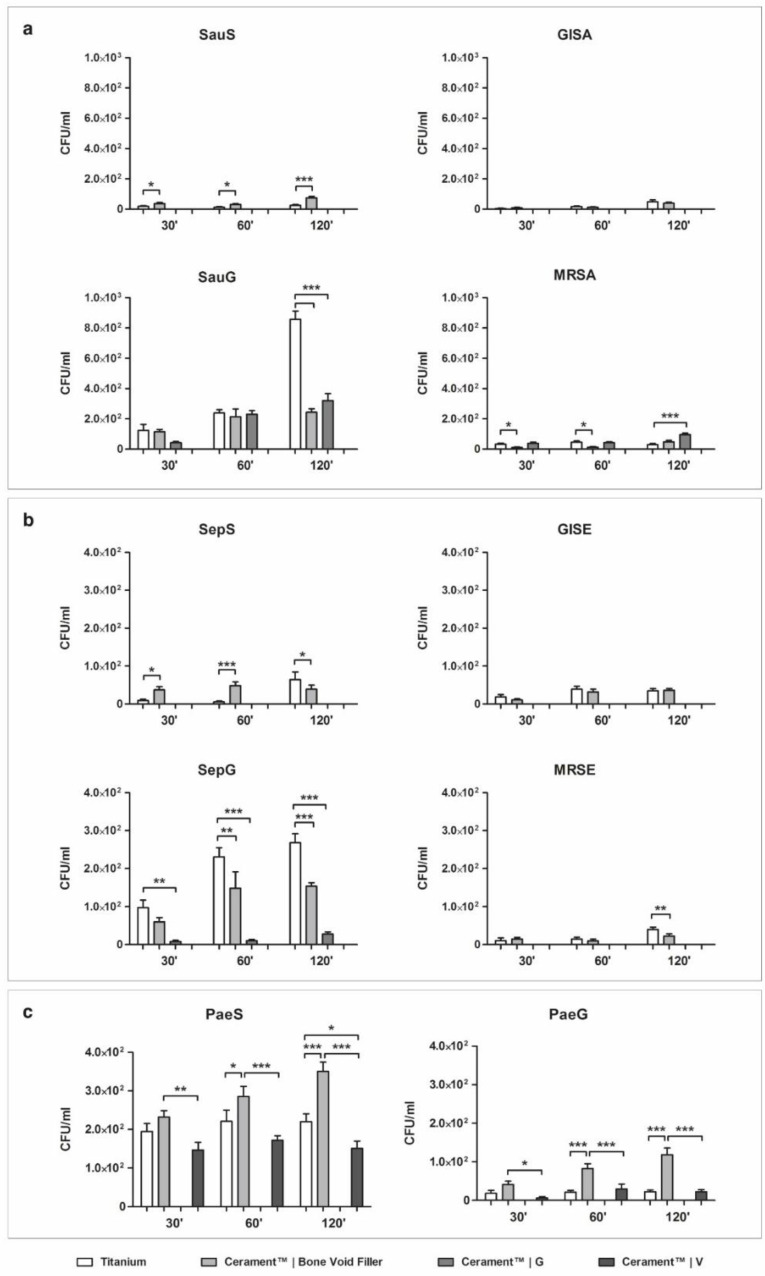
Bacterial adhesion on material surface as determined by plating and colony counting. (**a**) *Staphylococcus aureus*, (**b**) *Staphylococcus epidermidis* and (**c**) *Pseudomonas aeruginosa* adhesion on sandblasted titanium (ST), ceramic bone void filler (CBVF), ceramic filler containing gentamicin (CG) and ceramic filler containing vancomycin (CV) after 30, 60 and 120 min of incubation. Statistical significance: *p* < 0.05 (*), *p* < 0.01 (**) and *p* < 0.001 (***).

**Figure 2 ijms-21-09250-f002:**
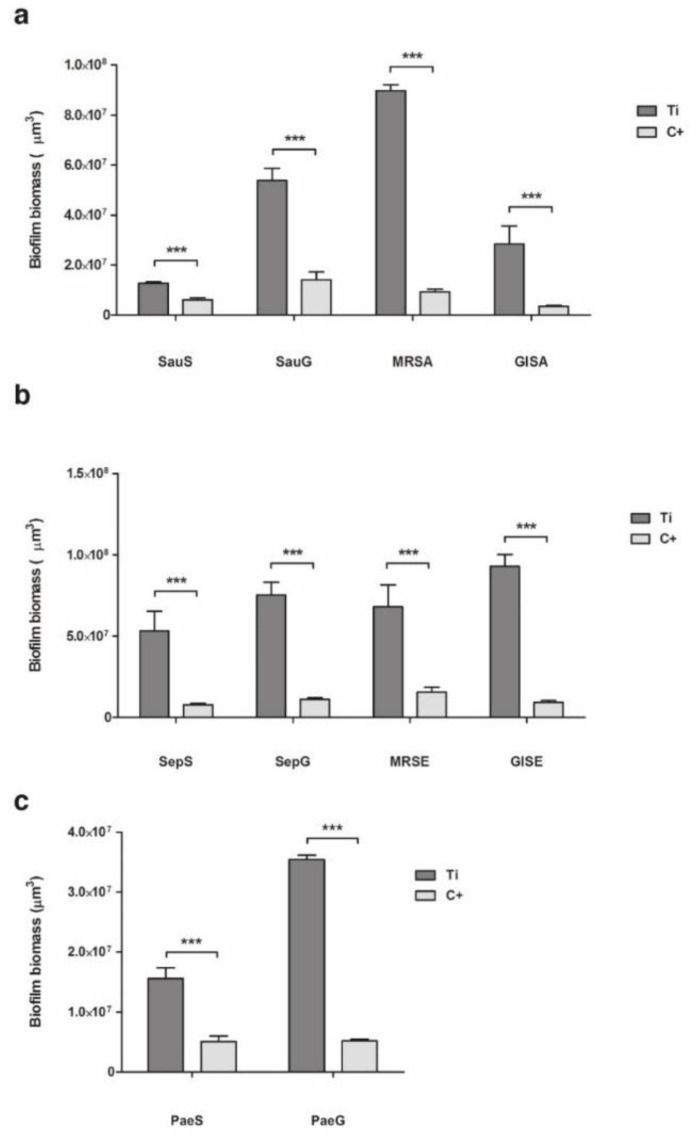
Biofilm biomass secreted by (**a**) *Staphylococcus aureus*, (**b**) *Staphylococcus epidermidis* and (**c**) *Pseudomonas aeruginosa* isolates on ST and CBVF. Statistical significance: *p* < 0.001 (***).

**Figure 3 ijms-21-09250-f003:**
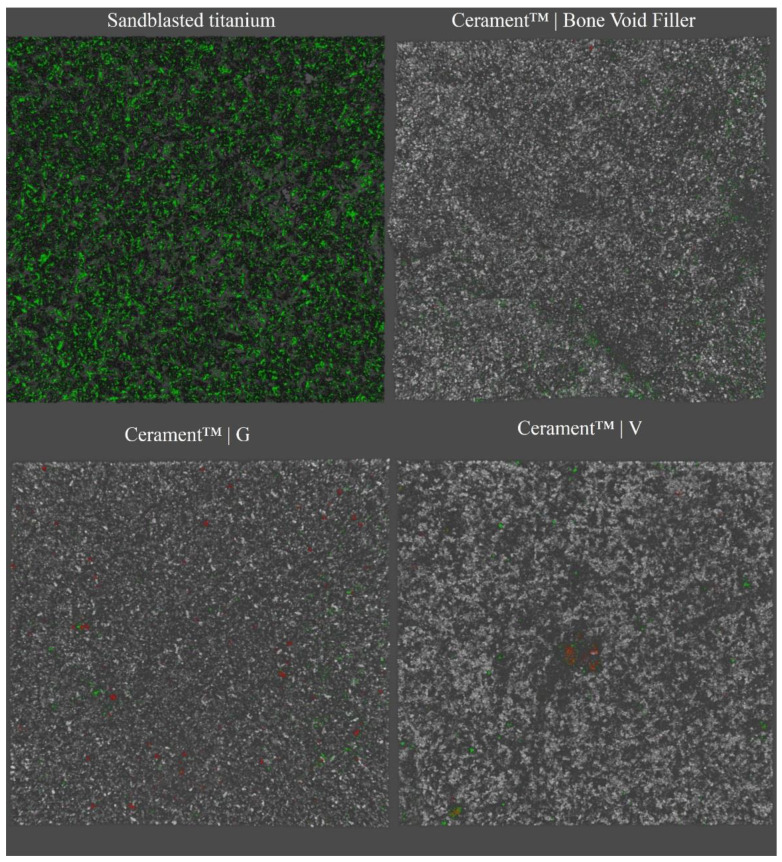
Representative confocal laser scan microscopy (CLSM) 3D images of sessile methicillin-resistant *Staphylococcus aureus* on ST, CBVF, CG and CV stained with Filmtracer™ LIVE/DEAD™ Biofilm Viability Kit. Magnification: 20×.

**Figure 4 ijms-21-09250-f004:**
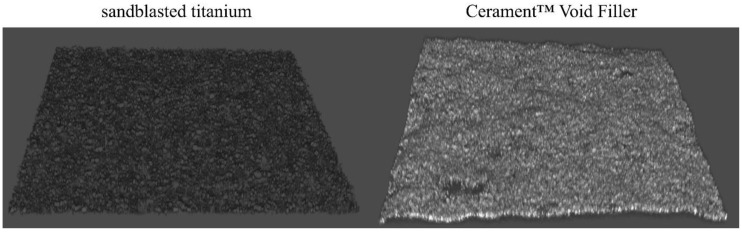
Representative CLSM 3D images of the surface characteristics of ST and CBVF. Magnification: 20×.

**Figure 5 ijms-21-09250-f005:**
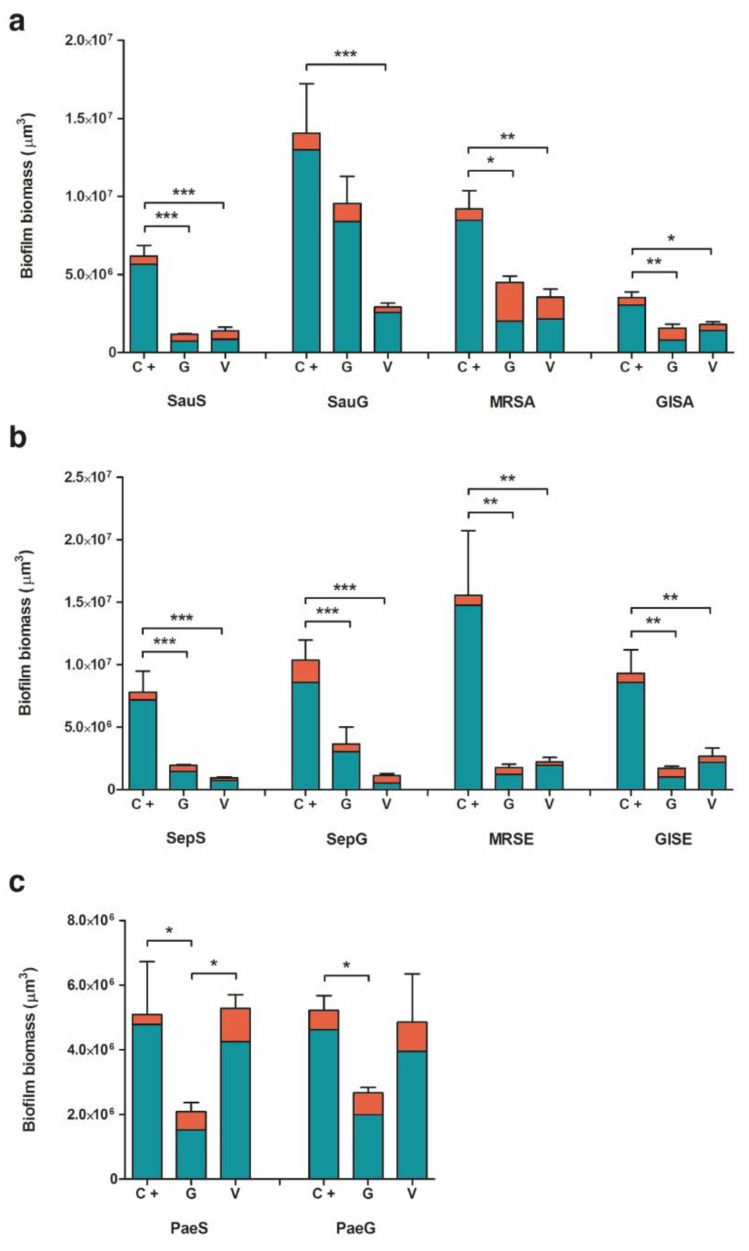
Ratio of live:dead cells (green:red) of (**a**) *Staphylococcus aureus*, (**b**) *Staphylococcus epidermidis* and (**c**) *Pseudomonas aeruginosa* isolates in biofilm formed on CBVF, CG and CV. Statistical significance: *p* < 0.05 (*), *p* < 0.01 (**) and *p* < 0.001 (***).

**Figure 6 ijms-21-09250-f006:**
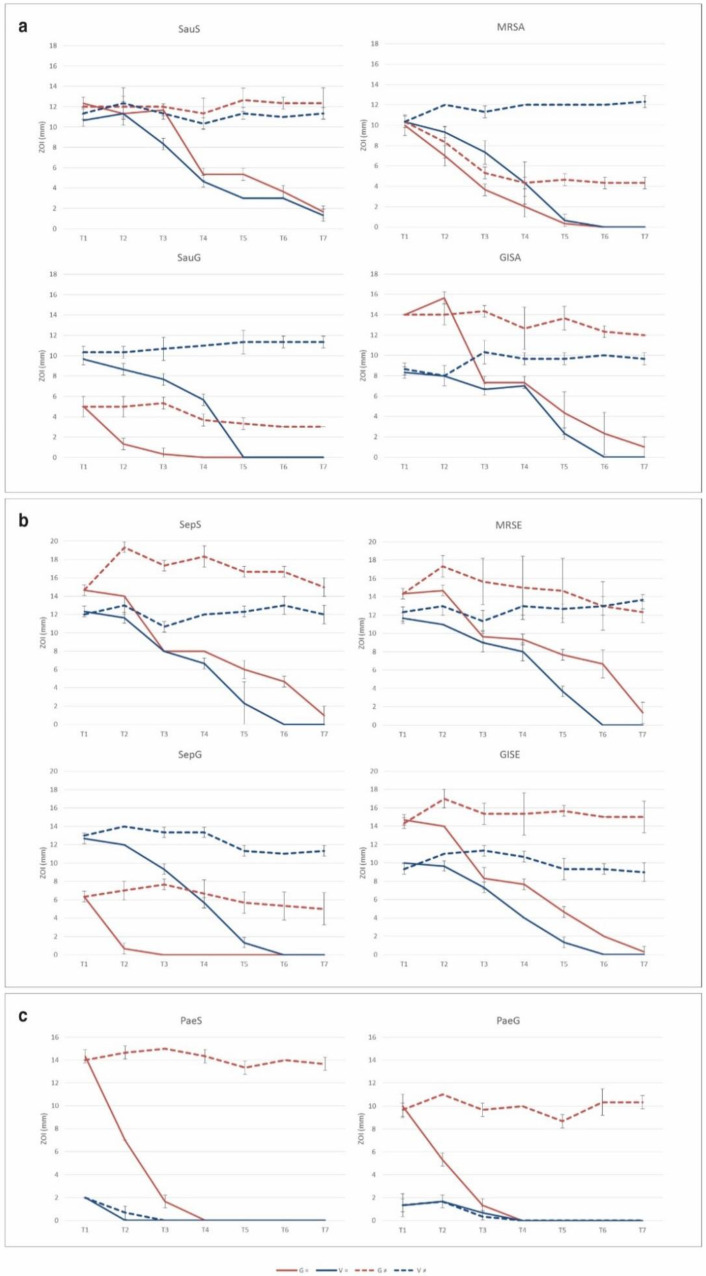
Zone of inhibition (ZOI) of (**a**) *Staphylococcus aureus*, (**b**) *Staphylococcus epidermidis* and (**c**) *Pseudomonas aeruginosa* growth on agar plates before (T0) and after (T7) exposure to CG/CV. ZOI (mm ± SD) are plotted against time (days). (=), exposure to the same CG/CV disc; (≠), exposure to a fresh CG/CV disc.

**Table 1 ijms-21-09250-t001:** Mortality rate by CLSM analysis. Data are expressed as percentage (%) of dead cells against the whole biofilm biomass. CBVF: Cerament^TM^ Bone Void Filler; CG: Cerament^TM^ G; CV: Cerament^TM^ V.

		CBVF	CG	CV
***S. aureus***	SauS	8.6	36.7	39.5
	SauG	7.7	12.1	11.6
	MRSA	7.9	55.3	38.6
	GISA	14.2	48.9	20.6
***S. epidermidis***	SepS	7.7	26.2	20.6
	SepG	15.8	16.8	52.3
	MRSE	5.0	31.2	12.4
	GISE	7.8	40.3	18.2
***P. aeruginosa***	PaeS	6.0	26.8	19.6
	PaeG	11.4	25.2	18.6

**Table 2 ijms-21-09250-t002:** Resistance profile of the tested strains to gentamicin before (T0) and after (T7) exposure to CG and after cycle on antibiotic-free medium (T14). Numbers indicate gentamicin concentrations (µg/mL). (=), exposure to the same CG disc; (≠), exposure to a fresh CG disc.

		T0	T7 (=)	T14 (=)	T7 (≠)	T14 (≠)
***S. aureus***	SauS	0.5	0.125–1	0.5	0.25–0.5	0.25
	MRSA	0.5	0.125–1	0.5	1	1
	SauG	256	256–512	256–512	512	512
	GISA	0.25	0.25–0.5	0.25	0.5–1	0.5
***S. epidermidis***	SepS	0.125	0.125–0.25	0.125	0.125–0.25	0.125–0.25
	MRSE	0.25	0.125–0.25	0.125–0.25	0.25	0.125–0.25
	SepG	128	256	128	128–256	128–256
	GISE	0.125	0.125	0.125	0.125	0.125
***P. aeruginosa***	PaeS	0.125	0.125	0.125	0.25	0.125–0.125
	PaeG	16	8	8–16	8	8–16

**Table 3 ijms-21-09250-t003:** Resistance profile of the tested strains to vancomycin before (T0) and after (T7) exposure to CV and after cycle on antibiotic-free medium (T14). Numbers indicate vancomycin concentrations (µg/mL). (=), exposure to the same CG disc; (≠), exposure to a fresh CV disc.

		T0	T7 (=)	T14 (=)	T7 (≠)	T14 (≠)
***S. aureus***	SauS	0.25	0.125–0.25	0.25	0.5	0.25
	MRSA	1	0.5–1	0.5	2	1–2
	SauG	0.5	0.5–1	0.5	1	0.5
	GISA	4	4	2–4	4	2–4
***S. epidermidis***	SepS	1	2	1	1–2	1
	MRSE	0.125	0.125	0.125	0.125	0.125
	SepG	1	1–2	1	2	2
	GISE	4	4–8	4	4–8	4
***P. aeruginosa***	PaeS	>1024	>1024	>1024	>1024	>1024
	PaeG	>1024	>1024	>1024	>1024	>1024

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
