# Peer review of "In Vitro Evaluation of Gentamicin or Vancomycin Containing Bone Graft Substitute in the Prevention of Orthopedic Implant-Related Infections"

_ijms, 2020, doi:10.3390/ijms21239250_

Round 1

Reviewer 1 Report

This is an excellent study which deals with a topic which has been widely discussed in the orthopaedic infection community in recent years. There have been small in vitro studies presented. You are to be congratulated on a strong methodology and elegant investigation of the problem. The paper is very well written and will be valued and understood by both laboratory scientists and clinicians alike.

My comments are just for clarification, rather than criticism:

1 The Introduction should be divided into paragraphs.

2 The order of the sections is odd. Methods should follow intro, then results, then discussion, then conclusions.

3 Can you give further information of the level of resistance of the PaeR strain which you chose. ?intermediate or not?

4 Line 343; is 24 hour culture long enough to find all the colony-forming units? Please explain

5 Line 345 Please write out the full name of each abbreviation (CLSM in this case) the first time you use it. The list of abbreviations at the end is helpful, but it is inconvenient to go to it during reading the paper.

6 In 'Results', line 77; 'resumed' should be 'summarized'

7 Line 136 needs modification; I would suggest: CSLM analysis allowed quantification of the biofilm biomass at ~6.2x106 ± 1.2x10 136 6 µm3

8 In 'Discussion', line 209; 'emerge' should be 'emergence'

9 Line 209; Vancomycin is not really an alternative to Gentamicin. It is an additional option. In PJI, many surgeons now add both to PMMA cement and your results would suggest that both are needed unless the micro is known. Even if we have preoperative biospies, we still cannot be certain of the intraoperative micro. Several studies have shown poor correlation between preop cultures and intraoperative samples.

10 Lines 270-272. You are correct that CG was used alone, but this was at least partly because CV was not available at the time of the study. It would be worth describing in more detail the outcome of this study with regard to the high effectiveness of CG, even when there were Gent resistant organisms cultured from the patients.

The figures are clear and greatly help to illustrate the results and significance.

Author Response

This is an excellent study which deals with a topic which has been widely discussed in the orthopaedic infection community in recent years. There have been small in vitro studies presented. You are to be congratulated on a strong methodology and elegant investigation of the problem. The paper is very well written and will be valued and understood by both laboratory scientists and clinicians alike.

The authors are pleased to read that the Reviewer #1 appreciated their paper and their work. Furthermore, we would like to thank the Reviewer #1 for the comments aimed at improving the quality of our manuscript.

My comments are just for clarification, rather than criticism:

1 The Introduction should be divided into paragraphs.

The authors would like to thank the Reviewer #1 for the suggestion and inform that the introduction section has been divided into paragraphs as recommended.

2 The order of the sections is odd. Methods should follow intro, then results, then discussion, then conclusions.

The authors agree with the Reviewer #1 that the order of paragraph in the paper is unconventional. However, we strictly followed the guidelines for paper preparation requested by International Journal of Molecular Sciences, which offers to authors a manuscript template to guide the formatting process. Since this is a journal requirement, we believe that we are not allowed to change the order of the paragraphs.

3 Can you give further information of the level of resistance of the PaeR strain which you chose. ?intermediate or not?

The authors thanks the Reviewer #1 for asking for this information. As reported in the manuscript, the gentamicin-resistant P. aeruginosa (PaeR) has a MIC value for gentamicin ≥ 16 and to vancomycin >1024.

To be thorough, here is reported the complete panel of tested antibiotic resulted from the Vitek2 analysis:

Antibiotic

MIC

Resistant/susceptible

Amoxicillin/clavulanic acid

≥32

R

Piperacillin/ tazobactam

32

R

Cefotaxime

≥64

R

Ceftazidime

32

R

Ertapenem

≥8

R

Imipenem

≥16

R

Meropenem

≥16

R

Amikacin

≤2

S

Gentamicin

≥16

R

Ciprofloxacin

≤0.25

S

Tigecycline

≥8

R

Colistin

≤0.5

S

Trimethoprim/sulfamethoxazole

160

R

4 Line 343; is 24 hour culture long enough to find all the colony-forming units? Please explain

The authors thank the Reviewer #1 for the interesting question. To evaluate the viability of the retrieved bacteria, the authors followed a protocol reported by Herigstad and colleagues and published on Journal of Microbiological Methods in 2001 [PMID: 11165341], which optimizes the count of viable bacteria suspended in a known volume. Briefly, the drop plate method by Herigstad et al. consists of distributing 10 µl of the sample in drops onto the designated quadrant of an agar plate. After the drops on the agar dried, the Authors suggest to invert and incubate Petri plates for 17–20 hours. According to our experience, a 24-hour culture is sufficient to count the number of colony-forming units when dealing with bacteria such as S. aureus, S. epidermidis, and P. aeruginosa.

5 Line 345 Please write out the full name of each abbreviation (CLSM in this case) the first time you use it. The list of abbreviations at the end is helpful, but it is inconvenient to go to it during reading the paper.

Due to the unconventional order of the paragraphs required by International Journal of Molecular Sciences, the results are reported before the materials and methods section. Hence, the acronym CLSM reported in line 345 is ok, since it appears for the first time in the text in line 122 (former line 120).

6 In ‘Results’, line 77; ‘resumed’ should be ‘summarized’

The authors thank the Reviewer #1 for the correction. According to the suggestion, the world “resumed” has been substituted with “summarized” in the revised version of the manuscript.

7 Line 136 needs modification; I would suggest: CSLM analysis allowed quantification of the biofilm biomass at ~6.2x106 ± 1.2x10 136 6 µm3

The authors agree with the Reviewer #1 that the sentence in ex line 136 should be modified. Hence, the authors revised the sentence according to this suggestion.

8 In 'Discussion', line 209; 'emerge' should be 'emergence'

Once again the authors thank the Reviewer #1 for the correction aimed at improving the quality of our manuscript. The authors inform the Reviewer #1 that the world “emerge” has been substitute with “emergence” in the revised version of the manuscript.

9 Line 209; Vancomycin is not really an alternative to Gentamicin. It is an additional option. In PJI, many surgeons now add both to PMMA cement and your results would suggest that both are needed unless the micro is known. Even if we have preoperative biospies, we still cannot be certain of the intraoperative micro. Several studies have shown poor correlation between preop cultures and intraoperative samples.

The authors agree with the Reviewer #1 that it is inexact to define vancomycin as the alternative antibiotic to gentamicin. For this reason, we modified the text in the discussion section reporting: “The increasing emergence of aminoglycoside-resistance lead to the development of new BGS formulation enriched with vancomycin, an additional option to gentamicin in the orthopedic field [29].”

Moreover, we agree with the Reviewer #1 that the a priori selection of the suitable product is a critical step especially in the treatment of fracture-related infections. Hence, we reported in line 313 the following sentence acknowledging the association of the two antibiotics in clinics: “Furthermore, the combined use of both gentamicin and vancomycin-eluting compounds might be considered for clinical cases presenting high-risk of infection recurrence due to polymicrobial contamination.”

10 Lines 270-272. You are correct that CG was used alone, but this was at least partly because CV was not available at the time of the study. It would be worth describing in more detail the outcome of this study with regard to the high effectiveness of CG, even when there were Gent resistant organisms cultured from the patients.

The authors thank the Reviewer #1 for pointing out this weak point of the discussion section. In accordance with this suggestion, this part has been modified as follows: “McNally and co-workers found a reduction of the recurrence rate of cOM of 4 % in a one-stage procedure compared to common reports in the literature of about 10% [21]. Briefly, they evaluated the use of CG in the management of deep bone infection in a one-stage procedure. Considering the unknown bacterial resistance pattern, the broader Gram-negative and Gram-positive spectrum, and the unavailability of CV at the time of the study, CG resulted effective in the treatment of chronic osteomyelitis, even in the presence of gentamicin resistant clinical isolates.”

The figures are clear and greatly help to illustrate the results and significance.

The authors are glad that the figures are clear and valid.

Reviewer 2 Report

Thank you very much for your nice research works.

My major is bone materials and dental implants, but your research theme is very similar to my field.

For brushing up your manuscript you should point out also the problems in dental region. Because in the oral cavity, the problems of bacteria are more severe, and your article is also useful to the reader in dental field.

I found some articles in oral surgery field saying the weakness of Cerament with dental implant for the bacteria contamination. PMID 31672257 or 30612838, by same research group. Your research show the usefulness of Cerament as the bone materials for dental region.

Some comments for M and M

This material is injectable material you wrote, correct?

if so, maybe it is difficult for making the specimen for bacteria adhesion and so on. It means difficulties for preparing the non-irregular surface for surface analysis. Please add the informations showing how to standardise the surface.

And for this type of study, sustained release test is important. Do you have the data for that? Or it has already published by someone? Long term sustained release ability is interesting and should be tested. In discussion, some comments for that should be added.

Author Response

Thank you very much for your nice research works.

My major is bone materials and dental implants, but your research theme is very similar to my field.

For brushing up your manuscript you should point out also the problems in dental region. Because in the oral cavity, the problems of bacteria are more severe, and your article is also useful to the reader in dental field.

I found some articles in oral surgery field saying the weakness of Cerament with dental implant for the bacteria contamination. PMID 31672257 or 30612838, by same research group. Your research show the usefulness of Cerament as the bone materials for dental region.

The authors would like to thank the Reviewer #2 for the positive comments and for the suggestions aimed at improving the quality of our manuscript.

The authors agree with the Reviewer #2 that there is the need to design future studies to investigate the effectiveness of Cerament formulations against dental pathogens such as P. gingivalis, Streptococci and enterococci and to enrich this promising compound with antibiotics locally used to treat dental infections (i.e. tetracycline, minocycline, etc.). Indeed, as reported in the articles indicated by the Reviewer #2, Cerament Bone Void Filler resulted inferior in terms of implant integration with respect to autologous bone, however we believe that the use of antibiotic-loaded Cerament might help hindering the deleterious effect of contamination of the oral implant by saliva-derived bacteria.

We are aware of the importance of dental infections, but, unfortunately, we decided to direct the study specifically at orthopedic infections, by choosing to test the efficacy of CERAMENTTM formulations against clinical isolates retrieved form patients with implant-related infections. Also the introduction section is totally oriented to the description of implant-related infections, reporting the incidence of pathogens and the use of antibiotic-containing bone graft substitutes in the treatment of orthopedic complications. This decision was based on the clinical use of CERAMENTTM formulations, which unfortunately are not used in the dental field yet. For the aforementioned reasons, we cannot discuss the usefulness of Cerament a bone material implant for dental region as suggested by the Reviewer #2.

Some comments for M and M

This material is injectable material you wrote, correct? If so, maybe it is difficult for making the specimen for bacteria adhesion and so on. It means difficulties for preparing the non-irregular surface for surface analysis. Please add the informations showing how to standardise the surface.

The authors confirm that all the tested CERAMENTTM formulations are injectable.

Furthermore, the authors agree with the Reviewer #2 that, when dealing with injectable materials, the standardization and the generation of discs with regular surfaces is crucial for the microbiological analyses. Therefore, as reported at the beginning of the paragraph 4.2. Evaluation of bacterial adhesion on material surface, the authors poured the freshly prepared injectable pastes in a custom-made mold of polydimethylsiloxane (PDMS). The PDMS mold was especially designed and created by the authors to standardize the quantity of CERAMENTTM and the shape of discs to be tested. Briefly, PDMS, a silicon-based, flexible, inert, and non-toxic polymer, was poured around titanium discs (6 x 2.5 mm) placed in a Petri dish. After an overnight incubation at 65 °C, the titanium discs were gently removed from the solidified PDMS, generating wells. The PDMS mold was then autoclaved before the use.

CERAMENTTM was then prepared following the manufacturer’s instruction and the obtained paste injected into the wells of the PDMS mold. Through the use of a spatula, the cement paste was flattened in each well to create discs with a surface as regular as possible. Once the CERAMENTTM formulations were dried out and solidified, cement discs were gently removed from the mold and conserved until analysis.

And for this type of study, sustained release test is important. Do you have the data for that? Or it has already published by someone? Long term sustained release ability is interesting and should be tested. In discussion, some comments for that should be added.

The authors would like to thank the Reviewer #2 for pointing out this important issue. Several in vitro and in vivo evaluations of CERAMENTTM formulations have been performed and reported in the literature and also the antibiotic release has been widely assessed. For instance, the in vitro release of gentamicin was tested and reported by Butini et al., 2018 [PMID: 29096369]. In this study, they reported an initial burst (within 24 hours) of gentamicin eluted from CERAMENTTM followed by a plateau in which lower concentrations of the antibiotic are released over a longer period. Furthermore, the release patter in vitro was compared with the results obtained by the release kinetics in vivo by Stravinskas and colleagues [PMID: 27678329].

Nonetheless, the long term behavior of CERAMENTTM compound was also investigated through the use of animal model to assess its mechanism of action and resorption in a setting closer to clinics [PMID: 32840717; Wang JS, et al., Biomechanics and bone integration on injectable calcium sulfate and hydroxyapatite in large bone defect in rat. 2006].
